# Using the Journey to Health and Immunization (JTHI) Framework to Engage Stakeholders in Identifying Behavioral and Social Drivers of Routine Immunization in Nepal

**DOI:** 10.3390/vaccines11111709

**Published:** 2023-11-10

**Authors:** Nicole Castle, Surakshya Kunwar, Leela Khanal, Lisa Oot, Katharine Elkes, Swechhya Shrestha, Anjali Joshi, Prasanna Rai, Sanju Bhattarai, Biraj Man Karmacharya

**Affiliations:** 1JSI Research & Training Institute, Inc., 2733 Crystal Drive, 4th Floor, Arlington, VA 22202, USA; lisa_oot@jsi.com (L.O.); katharine_elkes@jsi.com (K.E.); 2Department of Public Health, Dhulikhel Hospital-Kathmandu University School of Medical Sciences, Dhulikhel 45200, Nepal; anjalijoshi@kusms.edu.np (A.J.); prasannarai@kusms.edu.np (P.R.); birajmk@kusms.edu.np (B.M.K.); 3Independent Researcher, Kathmandu 44600, Nepal; leelakhanal2017@gmail.com; 4UNICEF Nepal, UN House, Pulchowk, Kathmandu P.O. Box 1187, Nepal; swshrestha@unicef.org (S.S.); sbhattarai@unicef.org (S.B.)

**Keywords:** behavioral science, Nepal, vaccination, vaccine acceptance, stakeholder engagement

## Abstract

Although the Government of Nepal has achieved high and sustained childhood vaccination coverage, reaching under-immunized and zero-dose children requires different approaches. Behavioral science offers promise in better understanding the drivers of vaccination and development of more effective programs; however, the application of behavioral science to immunization programs in Nepal is nascent. Through the Behavioral Science Immunization Network, JSI, UNICEF Nepal, and Dhulikhel Hospital–Kathmandu University School of Medical Sciences established a Behavioral Science Center to engage a diverse group of stakeholders in increasing the capacity of practitioners to use behavioral science in immunization programming. As a result of the engagement during formative research, government stakeholders requested and applied tools from behavioral science to solve different immunization challenges. Of particular value was the use of the Journey to Health and Immunization framework, which helped stakeholders identify behavioral and social drivers of zero-dose communities in Kathmandu. Our experience in Nepal demonstrates that there is strong demand for approaches and tools from behavioral science to use in relation to immunization and that this type of engagement model is effective for generating demand for and strengthening capacity to use behavioral science approaches.

## 1. Introduction

The Government of Nepal has prioritized its immunization program and has a goal of fully vaccinating 95 percent of children aged 12–23 months by 2030 [1,2]. The country’s immunization program is considered a success; compared to its peers, Nepal has achieved high and sustained vaccination coverage (80 percent in 2022) [2,3]. However, the percentage of children aged 12–23 months that did not receive any vaccinations increased from 1 percent in 2016 to 4 percent in 2022 [2]. These children are at higher risk of contracting vaccine-preventable diseases.

Behavioral science offers promise in understanding the drivers of vaccination and in developing more effective, people-centered health programs. Theoretical frameworks, including the Capability–Opportunity–Motivation–Behavior (COM-B) model and the Increasing Vaccination Model, have been used and adapted for global and country guidance on routine immunization and COVID-19 vaccination programs [4,5,6,7,8]. However, the use of these frameworks by practitioners in low- and middle-income countries, such as Nepal, for immunization programming is nascent [9]. 

In Nepal, multiple models of behavior change including Diffusion of Innovation Theory, Health Belief Model, COM-B, and Social Cognitive Theory have been applied to research and intervention design within the areas of HIV/AIDS; water, sanitation, and hygiene; mental health; and sexual and reproductive health [10,11,12,13,14,15]. The majority of the authors associated with these studies are from institutions in high-income countries such as Japan, Switzerland, the Netherlands, and the United States. Many studies have examined the socioeconomic determinants of vaccination and the demand- and supply-side barriers and enablers to vaccination in Nepal. However, none of the studies used a specific behavior change model or were related to intervention design [1,16,17,18,19]. 

As part of the Behavioral Science Immunization Network project, JSI Research & Training Institute, Inc. (JSI) explored different ways to increase the knowledge and strengthen the capacity of immunization practitioners to use behavioral science approaches in their programs. In Nepal, JSI wanted to learn what type of community of practice structure could support capacity strengthening and test different approaches to understand behavioral and social drivers of low routine vaccine uptake in specific communities. Based on a recommendation identified through a project scoping exercise, JSI, United Nations Children’s Fund (UNICEF) Nepal, and Dhulikhel Hospital–Kathmandu University School of Medical Sciences (DH-KUSMS) established a Behavioral Science Center (BSC) to bring together immunization and non-immunization stakeholders to apply behavioral science to immunization policy and programs. The BSC serves as a platform to employ practitioner-friendly behavioral models in formative research and intervention design and engage key stakeholders in this process to demonstrate the value of applied behavioral science and strengthen their understanding of the field.

## 2. Materials and Methods

Through the Behavioral Science Immunization Network project, the BSC had two primary objectives: (1) document the use of behavioral science tools and approaches in relation to immunization and (2) increase capacity of BSC members to apply behavioral science to immunization. To achieve these objectives, the BSC, in consultation with the Ministry of Health and Population (MOHP), conducted formative research on the social and behavioral drivers of childhood vaccination in select wards in Kathmandu Metropolitan City and Sudurpaschim and Madesh provinces. The formative research process facilitated capacity strengthening of the DH-KUSMS core team and other BSC members, enabled the team to document the use of behavioral science tools and approaches in relation to immunization, and supported stakeholder engagement and advocacy.

### 2.1. Selection of Practitioner-Friendly Behavior Change Model

As a first step, the team selected a practitioner-friendly behavior change framework to apply to the formative research process. The team decided to use the Journey to Health and Immunization (JTHI) framework, a model and journey mapping tool developed by UNICEF and its partners in 2017 [20,21]. UNICEF and partners developed the JTHI in response to the limitations of applying other behavior change theories such as the Health Belief Model to caregiver vaccination and health behaviors [22]. The JTHI enables practitioners to understand both caregiver and health worker journeys, providing valuable insights into their experiences and challenges across six essential domains (Figure 1) that affect caregivers and health workers before, during, and after vaccination services are delivered [20,23].

The team selected the JTHI framework over other models for a number of reasons. First, the JTHI draws from the socio-ecological model, a theoretical framework with which the team had experience [20]. Second, the JTHI was developed within the contexts of immunization and low- and middle-income settings [22,23]. For example, the JTHI has been adapted as a qualitative inquiry approach and applied as an analytical framework in multiple studies focused on low- and middle-income settings, including Sierra Leone, Nigeria, Democratic Republic of the Congo, and Mozambique, as well as several other Gavi-supported countries [23,24,25]. It has also been included in COVID-19 vaccination guidance developed by the United States Centers for Disease Control and Prevention [26]. Third, the JTHI is practice oriented, uses human-centered design (HCD) principles, supports problem prioritization, and can be used to design tailored solutions to improve outcomes for individuals and communities [20,22].

The team also consulted with BSC members to gather inputs related to existing challenges in vaccine uptake, identify priority populations to focus on in the formative research, and provide feedback on the formative research proposal, including the selection and use of the JTHI. The BSC members agreed that the JTHI was helpful for Nepal’s context and would support the MOHP’s objective of fully vaccinating all children by identifying the behavioral barriers faced by caregivers with children who have not received any vaccinations or only some vaccinations, as well as their respective service providers.

### 2.2. Stakeholder Engagement throughout the Formative Research Process

Throughout the formative research process, the core KUSMS team consulted with BSC members, which included representatives from government agencies (e.g., Nepal Health Research Council (NHRC); National Health Education, Information, and Communication Center (NHEICC); National Health Training Center (NHTC); MOHP Family Welfare Division; provincial and municipal health directorates), nongovernmental organizations (e.g., HERD International, UNICEF), academic institutions (e.g., Tribhuvan University, Patan Academy of Health Sciences), and a professional society (Nepal Public Health Association) to ensure high levels of engagement. The team also held several consultative meetings with ward-level stakeholders, some of whom were members of the BSC and others who were not. The team leveraged these consultative meetings to demonstrate the value of using behavioral science tools and increase knowledge of behavioral science while employing practitioner-friendly behavior models to improve immunization practice. 

These consultative meetings provided an opportunity for the core DH-KUSMS team to introduce the JTHI and the importance of taking a human-centered perspective. The core team shared their intention to leverage previously unutilized tools to determine which approaches were effective and ineffective in Nepal’s context. Additionally, the consultative meetings provided an opportunity for BSC members to provide feedback on the formative research design, discuss behavioral science tools, discuss formative research findings, including how they were used to inform the design of interventions, share updates and learnings, and commit to future activities for the BSC to take forward. Figure 2 demonstrates stakeholder engagement throughout the formative research process.

### 2.3. Learning by Doing: Formative Research as Capacity Strengthening

#### 2.3.1. Overview of Formative Research

Formative research was conducted using rapid inquiry in selected wards of Bagmati, Madesh, and Sudurpaschim provinces to understand the social and behavioral drivers influencing routine vaccine uptake. The study sites were selected purposively through consultation with BSC members and stakeholders from all levels of government and based on vaccination coverage data, ecological representativeness, and inclusion of urban poor, underserved, and marginalized communities. Caregivers and health workers, including female community health volunteers (FCHVs), were selected purposively. We conducted in-depth and key informant interviews with 52 caregivers (i.e., mothers, fathers, and/or grandparents of children between 6 months and 24 months of age) and 12 health workers and FCHVs. The sample size was determined based on the mean and code saturation. The BSC team adapted the Behavioral and Social Drivers of Vaccination (BeSD) guide using HCD tools and the JTHI to develop the interview guides. Trained research assistants facilitated the data collection, maintaining ethical consideration. The study was approved by the Institutional Review Board of the Nepal Health Research Council (reg. no. 294/2022, approved on 18 July 2022) and the Institutional Review Committee of Kathmandu University School of Medical Sciences (approval no. 150/22, approved on 4 August 2022). Only participants who provided informed consent were included in the study.

Some challenges encountered during the formative research included language barriers, participant availability, and gender-related barriers. To overcome language barriers, the research team included research assistants proficient in Hindi, Bhojpuri, and Maithali languages. Because the majority of participants were daily-wage workers or laborers, there was limited opportunity to conduct interviews during the day. In response, our team, with support from FCHVs, made frequent visits to accommodate work schedules. In certain communities, strict gender norms restricted females from interacting with male interviewers or even in participating in the study without their male partner’s consent. To address this, the research team worked with FCHVs to obtain consent and conducted interviews with female caregivers using female research assistants and with male caregivers using male research assistants, ensuring cultural sensitivity and cooperation.

#### 2.3.2. Capacity Strengthening

The project’s scoping exercise and consultative sessions with stakeholders found that there was limited capacity in applied behavioral science and the use of practitioner-friendly models for immunization programs, which required capacity development to become a core element for applying behavioral science within the context in Nepal. Thus, the formative research process itself became a capacity-strengthening activity. Both the core DH-KUSMS team and BSC members participated in capacity strengthening throughout the formative research process. 

JSI and UNICEF Nepal used a “learning-by-doing” approach for the core DH-KUSMS team. First, the team participated in a 2-day workshop on the approaches and processes that would be used while conducting formative research. This included an overview of the JTHI, behaviorally informed key informant interview tools, and field data collection methods from anthropology including kuragraphy and observation. Kuragraphy is an informal, unstructured conversation with community members [27,28]. The workshop was followed by a 3-day training course on the HCD process, which was used in both the formative research and intervention design and included a focus on facilitating co-creation among multiple stakeholders. 

After initial training sessions, JSI and UNICEF Nepal provided on-site coaching to the DH-KUSMS team as they conducted formative research, including coaching on pre-testing tools and performing a preliminary analysis of findings. The team used the JTHI to analyze formative research findings. Applying the JTHI to a real-life scenario helped increase the DH-KUSMS team engagement and interest in the process, serving both as a means for learning and as actual analysis and use of the formative data.

In addition to the activities associated with the formative research process, we conducted training on HCD, social and behavior change approaches, and interpersonal communication (IPC) for the DH-KUSMS team, government agency representatives, other academics, health workers, and FCHVs. The training provided an overview of practitioner-friendly behavior change models, including the Behavioral Drivers Model and socio-ecological model [29,30], the HCD process, tools, and its relationship to behavioral science, and the role of interpersonal communication (IPC) in providing respectful care. 

## 3. Results

Engaging stakeholders through an academia-hosted network was an effective way to demonstrate the value of applied behavioral science in relation to immunization and generate stakeholder buy-in and commitment. Application of the JTHI and engaging stakeholders throughout the process facilitated further application of related tools and supported evidence-based advocacy for applying behavioral science to immunization programs.

### 3.1. Application of Tools by Government Stakeholders

The DH-KUSMS, JSI, and UNICEF Nepal team was one of the first groups to introduce and apply the JTHI tool to immunization challenges in Nepal. Engaging government stakeholders in the selection and application of the JTHI and in sharing the formative research findings using the JTHI as an analysis framework encouraged the government to apply the tool to other challenges. For example, the MOHP requested that the DH-KUSMS research team support the government’s response to a measles outbreak in Nepalgunj, Banke district of Lumbini province using the JTHI and rapid inquiry approaches. Through applying these methods, the DH-KUSMS team was able to quickly identify specific behavioral and social drivers of under-immunization that contributed to the outbreak in wards 5, 7, 8, 9, and 11 of Nepalgunj Sub-Metropolitan City (Table 1). 

Based on the rapid inquiry findings, the DH-KUSMS team worked with municipal and ward-level health authorities to design tailored strategies to encourage eligible children to receive the measles vaccine during the outbreak response. These strategies included engaging religious and ward leaders and community influencers; engaging male family members in vaccination counseling; door-to-door vaccination; and advocacy to revise outbreak response guidelines to be inclusive of children with different abilities, leading to implementation of door-to-door vaccination and travel cost reimbursement for families taking children with different abilities to health facilities for vaccination, ensuring equitable access.

Additionally, at the same time as DH-KUSMS was conducting its formative research, the NHRC, with support from UNICEF Nepal, conducted a study on health-seeking behaviors using similar approaches and the JTHI as an analytical framework [31]. These two teams applying the JTHI concurrently supported capacity strengthening and knowledge sharing among stakeholders in the BSC. Further, the NHRC application of the JTHI demonstrated its value and effectiveness for programs outside of immunization, including maternal health (e.g., institutional delivery) and nutrition. As a result of engagement with the MOHP and the NHTC, the team incorporated the JTHI into the government training curriculum and received accreditation within the government system. To date, more than 500 participants from various government departments, partner organizations, and Kathmandu University have been trained. 

### 3.2. Evidence-Based Advocacy and Use of Behavioral Insights

The team conducted formative research using the JTHI to generate evidence of vaccine acceptance demand gaps and shared the findings through the BSC with relevant stakeholders and policymakers. This allowed us to conduct evidence-based advocacy, advocating for context-specific demand programs that are tailored to the needs of underserved communities and those in the last mile. For example, after sharing the research findings analyzed using the JTHI (Figure 3) with the Kathmandu Municipality Health Department, the department held an emergency session to identify and immunize communities located in the last mile, promptly using behavioral insights to implement new programs. After revisiting the same community for implementation research, the team found that many community members reported their children had received vaccines with the assistance of the government team. Moreover, the findings from the formative research were used to advocate at the policy level during the National Immunization Strategic Planning workshop.

In addition to using evidence to advocate for demand programs, it was used to advocate for further application of and capacity strengthening in behavioral science. For example, after sharing formative research results, the Department of Health Services requested that the BSC support health promotion and education officers incorporate behavioral and social science theories and models into their practices to better address health issues within their communities. Additional recommendations that resulted from the BSC’s stakeholder engagement and advocacy were to integrate behavioral science processes and approaches into existing provincial-level rapid-response teams’ training and guidelines and to expand the use of processes such as HCD to all levels of government to improve service delivery. Moreover, this advocacy has resulted in a consortium of public health universities in Nepal discussing incorporating applied behavioral science into graduate programs’ curriculum. 

## 4. Discussion

Our experience in Nepal demonstrates that there is strong demand for approaches and tools from behavioral science to use for immunization and other health programs and that bridging the gap from theory to practice is achievable. We found the JTHI framework to be a practitioner-friendly model that can be used at the subnational level to identify issues faced both by health providers and caregivers across multiple levels of the health system. The JTHI served as a useful framework for the design of formative research and analysis of data and helped to identify limitations within the current immunization program to meet the needs of marginalized populations in Nepal (i.e., urban poor, rural–remote communities). 

In order to apply lessons learned from the formative research, we found that a strong evidence-based framework was critical, as was buy-in and support from policymakers, research institutions, and immunization practitioners regarding the application and use of behavioral science tools and methods. These findings are aligned with the dimensions included within the Context and Implementation of Complex Interventions framework [32]. At the start of the project, we learned that there was limited expertise among these cadres, and specifically among immunization practitioners, on the application and use of behavioral models and found it imperative to provide orientation on applied behavioral science and how it can be integrated into programs via different mechanisms. 

In addition to orientation on behavioral science methods and tools, applied learning (learning by doing) was essential to the success of our approach. Others have identified “learning by doing” as an effective strategy to strengthen capacity and develop new skills [33,34,35]. Practical application of the JTHI framework by stakeholders was essential for them to understand its application and use and to advocate for the approach to be utilized in subsequent immunization programming (and programming outside of immunization). 

To institutionalize this type of application and use of behavioral science models, methods, and tools, our experience showed that there is value in forming networks to develop capacity in and institutionalize the use of behavioral science. However, not surprisingly, the context and structures through which these networks are facilitated matter. In Nepal, situating a network within an academic institution has demonstrated potential as an effective model for capacity development and application of behavioral science in immunization and broader health programming. In fact, other universities in different contexts have successfully created dedicated centers or communities of practice (COPs) to enhance learning and instruction of specific academic topics. One study found that Science, Technology, Engineering, and Math (STEM) instructors who were part of a STEM COP were more likely to use student-centered approaches to teaching, such as engaging students in discussion and asking questions. Students on courses taught by instructors who were members of the COP showed higher levels of active learning than those in courses taught by instructors who did not participate in the COP [36]. Another study from Central Queensland University (CQU) in Australia noted that institutions are increasingly using COPs to improve staff work performance, knowledge exchange, research outcomes, and more [37]. Furthermore, programs in Uganda and South Africa have also demonstrated the benefits of working through academic institutions to strengthen capacity in behavioral science [38,39]. These programs also worked to develop capacity through practical application of skills to address public health issues and underscore the benefits of integrating capacity development through applied research and response to government needs. 

In order to scale and sustain the capacity and use of behavioral science for public health programming, it is essential to have an institution pushing a behavioral science agenda and serving as the “go to” entity for support within the country. We found that building local capacity to create and then institutionalize the BSC was essential to its success and sustainability. Our findings echo those of Stevens et al. [40], who found that in order to create a successful and sustainable community of practice you need to: (1) gather a core group of motivated individuals, (2) keep participants caring and engaged, and (3) delegate as much as possible to the local institutions as soon as you can [40]. In development of the BSC, we found a core group of individuals invested in practical application of behavioral science and aligned to meet government research and response needs, helping to translate research into policy and programs and demonstrating the value of applied behavioral science. DH-KUSMS took on ownership of the COP early on in the process, establishing a physical location within Kathmandu University and working to build a team of experts within the University to respond to government research and program requests. 

As part of its capacity development strategy, the BSC is in the process of incorporating a behavioral science curriculum for its Masters of Public Health (MPH) students in an effort to institutionalize knowledge and develop capacity among public health professionals early in their career. There is potential to scale this curriculum across Nepal because of KUSMS’ linkages to other public health institutions in the country. This strategy is similar to approaches other academic and training institutions have taken to strengthen health professionals’ knowledge, skills, and practices. The Kenya Medical Training College revised the Expanded Program on Immunization curriculum to include content on Effective Vaccine Management to strengthen pre-service training. This helped improve vaccine management practices and develop a larger pool of professionals with understanding of Effective Vaccine Management [41]. The University of North Carolina at Chapel Hill incorporated content on applied implementation science in its MPH curriculum in response to the need to improve the skills of future public health professionals to design, implement, and evaluate programs that are responsive to local and global challenges and the competencies needed to realize the vision of Public Health 3.0 in the United States. The curriculum is designed to equip MPH students with the skills needed to apply implementation science in different contexts [42]. Incorporating behavioral science content into pre-service training for health practitioners is a potentially sustainable strategy to strengthen capacity. We recommend countries explore integrating behavioral science tools and methods (e.g., HCD, JTHI) as part of a pre-service curriculum so that health professionals are confident in and capable of using those tools, understanding the data that are gathered through the participatory processes, and designing locally tailored solutions to problems. 

In our work, we also found that mass media and traditional research and expert-driven approaches are prioritized instead of user-centered approaches. The application of HCD approaches to identify barriers to vaccine uptake among urban poor communities in Nepal provided invaluable insights that could not have been captured without an iterative, participatory process. In addition, this approach has led to changes among BSC members in the way they are thinking about immunization service delivery. Members are looking at persistent challenges within the health system and are asking to use applied behavioral science to address them. For example, one member asked: “How can we make health services available 24 h a day? To do this, we need to examine behaviors of the health system and the health provider to understand what the barriers are and break them down into manageable pieces to address.” This is a reflection of the benefits of using participatory processes such as HCD to address complex real-world challenges, including increased relevance to context, better translation of research to action, and potential to lead to solutions that are more readily adopted and more effective [43,44]. Thus, while the application of participatory methods can be more expensive and time intensive than other strategies, we contend it is worth the investment. We recommend that countries consider how to integrate participatory processes into pre-service training, not just for immunization, but for the benefit of broader health service policy, planning, and delivery. The integration of HCD into training curricula was also noted by Chen et al. as having an implication for public health research and practice [44].

## 5. Challenges and Limitations

Our project faced several challenges and limitations. It took several months, as well as a considerable initial financial and human resource investment, to establish and build the capacity of the BSC at DH-KUSMS. Additionally, the duration of the JSI program was short, less than 12 months, limiting the duration of program implementation and measurement post implementation. The short time-frame limited our ability to provide on-going organizational and technical capacity strengthening to the BSC as the BSC itself worked to become self-sustaining. Additional time and financial resources are still needed to strengthen the technical, operational, and financial skills of the BSC to develop and respond to ministry and funder requests for programming. Ideally a phased technical assistance approach over a three-to-five-year period would have been put in place to support DH-KUSMS and their efforts to respond to ministry and funder requests for research and programming. 

## 6. Conclusions

The JTHI is a practical framework that can be used to improve understanding of immunization services and outcomes and is particularly useful at subnational levels where localized issues can be identified and addressed. Our work in Nepal has also demonstrated that the JTHI model can be used as a practical model to design formative research, and it can also be used as an advocacy tool to improve immunization programming. As applied behavioral science is new for many immunization practitioners, orientation and capacity development will be required in many countries. Once capacity is developed within countries, as the work in Nepal has demonstrated, local institutions will be able to apply behavioral science tools and methods to address global immunization priorities (i.e., reaching zero-dose children, addressing norms and behaviors that prevent vaccine uptake). 

In order to expand application and use of behavioral science in countries, networks similar to the BSC that was developed in Nepal will be critical. We believe the BSC model can be applied in other countries if the following are in place: (1) support for orientation and advocacy for behavioral science application, (2) an existing institution with the interest to establish and build a behavioral science center, and (3) technical and operational assistance to the institution that can help practitioners respond to research and program requests. We believe that the best way to support this type of process is a phased approach of technical and operational assistance over a period of three to five years.

## Figures and Tables

**Figure 1 vaccines-11-01709-f001:**
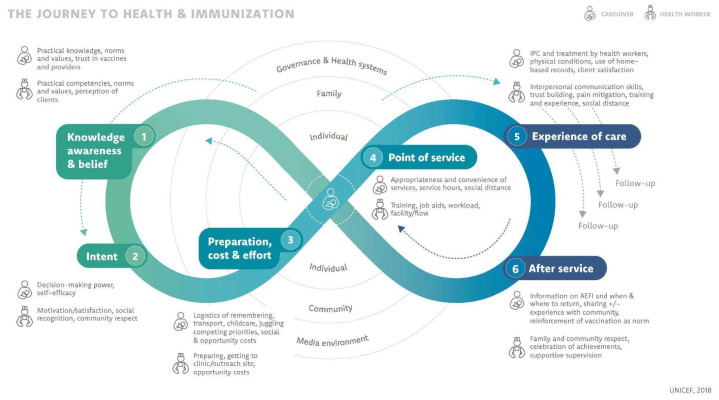
Journey to Health and Immunization [21].

**Figure 2 vaccines-11-01709-f002:**
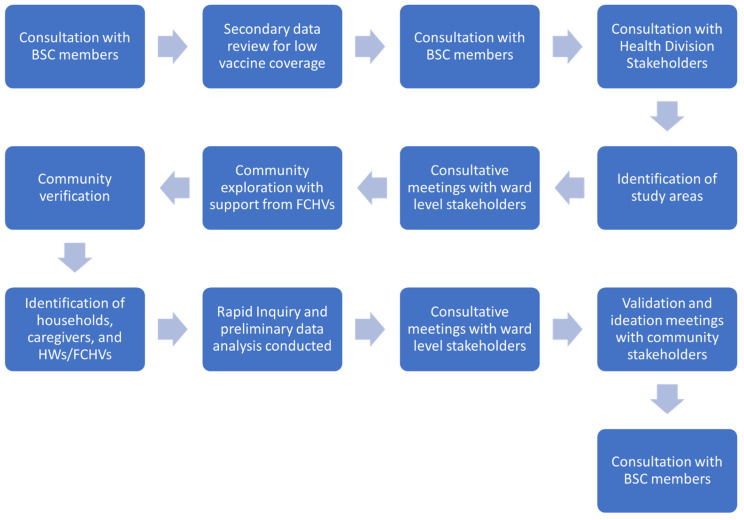
Stakeholder engagement throughout the formative research process.

**Figure 3 vaccines-11-01709-f003:**
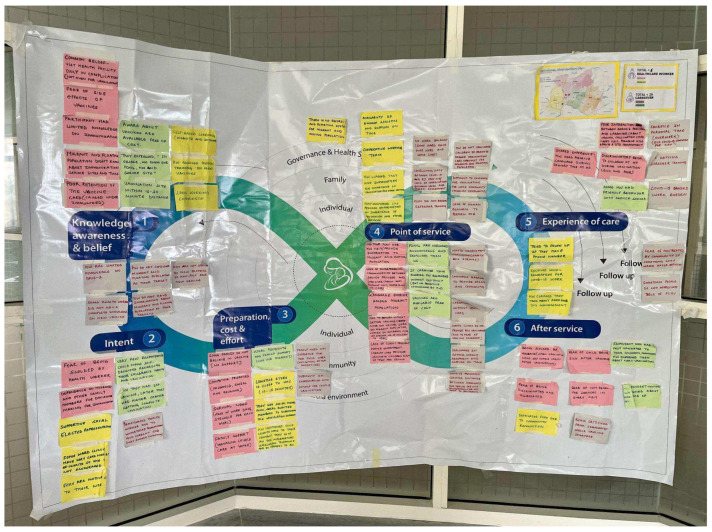
Journey to Health and Immunization map for Kathmandu Metropolitan City.

**Table 1 vaccines-11-01709-t001:** Social and behavioral drivers of under-vaccination for measles from caregiver perspective according to JTHI domains.

JTHI Domain	Barriers	Enablers
Knowledge, Awareness, and Belief	Unaware of importance of vaccine cardBelief that the measles vaccine causes infertility later in life Fear of side effects because of past experiences with AEFI	Aware of importance of vaccineKnowledge of where to take children for vaccinationWard representatives, relatives, and FCHVs shared information about the vaccine
Intent	Caregivers could not take children because they had to workMale caregivers not allowing wives to bring children to vaccination	Caregivers received counseling on measles vaccinationFemale caregivers having decision-making power
Preparation, cost, and effort	Female caregivers do not have anyone to help with household work if they go to vaccinate their child	Health facility offering vaccination is in walking distanceLocal transportation is available
Point of service		Short wait times at health facilityHWs speak local languages
Experience of care	Caregivers experience rude treatment from HWs when they visited facility for ANC	HWs inform them about side effects, next date for vaccination, importance of vaccine card
After service	Fear of husband scolding female caregiver if child cries at night from side effects after vaccination	HWs provide information about next immunization visit

## Data Availability

No new data were created or analyzed for this study. The data from the project are available on request from the corresponding author.

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
