# Peer review of "Using the Journey to Health and Immunization (JTHI) Framework to Engage Stakeholders in Identifying Behavioral and Social Drivers of Routine Immunization in Nepal"

_vaccines, 2023, doi:10.3390/vaccines11111709_

Round 1
Reviewer 1 Report
Comments and Suggestions for Authors
Comments
This study used the Journey to Health and Immunization (JTHI) framework to engage stakeholders in identifying behavioral and social drivers of routine immunization in Nepal. The manuscript is interesting, however, lacks some important information in the main text. Here are some points I would like the authors to consider to further highlight the contribution of the study.
1. The introduction section of the article should show the main research question and purpose. Currently it is unclear.
2. In the methodology section of the article, the two primary objectives of BSC do not require two separate lines.
3. Line 77, “The Through”. Two words with capitalized first letters are wrong.
4. Manuscript requires ethical review information.
5. How can the results section adequately support the findings and conclusions of the study when there are no pictures or tables with the data presented?
6. Please add a Stakeholder engagement throughout the formative research process diagram or figure into the result section.
7. Final discussion section should include limitations and conclusions.
Author Response
Thank you for your review and feedback. Please see responses to each of your points below:
- The introduction has been revised (see highlighted text) to better demonstrate the main purpose.
- Formatting has been revised in the methodology section
- The sentence has been revised to read “Through the….”
- We have added ethical review information to the manuscript in the Methods and Materials section in section 2.3
- In the results section we added an image of the JTH map for Kathmandu Metropolitan City and a table related to the application of JTH to a measles outbreak.
- Process diagram added to materials and methods section in section 2.2
- Discussion section updated to clearly include sections on limitations and conclusions
Reviewer 2 Report
Comments and Suggestions for Authors
Major issues
1. The discussion should be rewritten in deep. The authors should Introduce more references in the discussion. There are only 3 references in the discussion, in my opinion this is insufficiente for a scientific publication. Your finding should related with the existent knowledge.
2. Write a paragraph or a sentence in the discussion about the limitations of your study.
Minor issues
1. “The Through the Behavioral Science Immunization Network project, the BSC had two primary objectives.” This sentence is hard to read it should be rewritten.
2. Please explain the “six essential domains that affect caregivers
3. The paper mentions that 80% of the children have been vaccinated by 2022. However, it then says that only 52% of children are vaccinated on time. This could potentially be misleading. Clarifying what “on time” means or providing more context would help maintain consistency. I suppose it may be mean within the vaccine schedule.
4. The material and methods is incomplete more explicit details should be provided eg how exactly the formative research was conducted – for instance, sample size, methods of data collection (e.g., surveys, interviews), criteria for selection of wards in the cities mentioned, and any challenges encountered.
5. Also in material and methods “kuragraphy,” and other methods are s mentioned without context or explanation. Write a brief description of these methods.
Comments on the Quality of English LanguageThe paper should be reviewed for clarity, eg.
"Of particular value was use of the Journey to Health and Immunization framework..." change into : "Of particular value was the use of the Journey to Health and Immunization framework..."
Change "Regarding routine immunization, multiple studies have examined the socioeconomic determinants of vaccination as well as the demand and supplyside barriers and enablers to vaccination in Nepal, but none of the studies used a specific behavior change model or were related to intervention design" into "Many studies have examined the socioeconomic determinants of vaccination and the demand and supply side barriers and enablers to vaccination in Nepal. However, none of the studies used a specific behavior change model or were related to intervention design."
This sentence is also hard to read "“The Through the Behavioral Science Immunization Network project, the BSC had two primary objectives.” it should be rewritten.
Author Response
Thank you for your feedback. Please see responses below:
Major revisions:
- We have revised the discussion section and introduced more references, more clearly showing our findings relation to existing knowledge.
- We have revised the discussion section to include a limitations section.
Minor revisions:
- Sentence has been rewritten.
- Added figure 1 of journey to health and immunization which shows the six domains.
- Sentence removed.
- Revised material and methods section to give detail about how the formative research was conducted.
- Added definition of kuragraphy in materials and methods section.
Regarding the quality of English language comments: Suggested revisions have been made for clarity and highlighted in the revised manuscript.
Round 2
Reviewer 1 Report
Comments and Suggestions for Authors
None.
Reviewer 2 Report
Comments and Suggestions for Authors
The authors have improved all the questions, that I have formulated. THe manuscript has improved a lot.